# Synthesis and Characterization of MIPs for Selective Removal of Textile Dye Acid Black-234 from Wastewater Sample

**DOI:** 10.3390/molecules28041555

**Published:** 2023-02-06

**Authors:** Maria Sadia, Izaz Ahmad, Zain Ul-Saleheen, Muhammad Zubair, Muhammad Zahoor, Riaz Ullah, Ahmed Bari, Ivar Zekker

**Affiliations:** 1Department of Chemistry, University of Malakand, Chakdara 18800, Lower Dir, Khyber Pakhtunkhwa, Pakistan; 2Department of Biochemistry, University of Malakand, Chakdara 18800, Lower Dir, Khyber Pakhtunkhwa, Pakistan; 3Department of Pharmacognosy, College of Pharmacy, King Saud University, Riyadh 11451, Saudi Arabia; 4Department of Pharmaceutical Chemistry, College of Pharmacy, King Saud University, Riyadh 11451, Saudi Arabia; 5Institute of Chemistry, University of Tartu, 14a Ravila St., 50411 Tartu, Estonia

**Keywords:** adsorption, acid black-234 dye, environment, selectivity

## Abstract

Herein, a molecularly imprinted polymer (MIP) was prepared using bulk polymerization and applied to wastewater to aid the adsorption of targeted template molecules using ethylene glycol dimethacrylate (EGDMA), methacrylic acid (MAA), acid black-234 (AB-234), 2,2′-azobisisobutyronitrile (AIBN), and methanol as a cross linker, functional monomer, template, initiator, and porogenic solvent, respectively. For a non-molecularly imprinted polymer (NIP), the same procedure was followed but without adding a template. Fourier-transform infrared spectroscopy (FT-IR), scanning electron microscopy (SEM), and a surface area analyzer were used to determine the surface functional groups, morphology and specific surface area of the MIP and NIP. At pH 5, the AB-234 adsorption capability of the MIP was higher (94%) than the NIP (31%). The adsorption isotherm data of the MIP correlated very well with the Langmuir adsorption model with Qm 82, 83 and 100 mg/g at 283 K, 298 K, and 313 K, respectively. The adsorption process followed pseudo–second-order kinetics. The imprinted factor (IF) and Kd value of the MIP were 5.13 and 0.53, respectively. Thermodynamic studies show that AB-234 dye adsorption on the MIP and NIP was spontaneous and endothermic. The MIP proved to be the best selective adsorbent for AB-234, even in the presence of dyes with similar and different structures than the NIP.

## 1. Introduction

The annual productivity of dyes is estimated to be one million tons. Textile industry produces large amount of dirty effluent through dyeing, washing, and other procedures. Synthetic dyes are the most dangerous compounds in wastewater because they are frequently made synthetically and have intricate aromatic structures that demonstrate light, oxidation, heat, and water stability. Dyes induce a variety of conditions, including cancer, allergies, mutation, dermatitis, and skin irritation. Therefore, removing dyes and other pollutants from the environment is critical for preventing contamination [1]. The quantity of dyes released into the water, on the other hand, prevents deoxygenating capacity and sunshine; therefore, aquatic life and biological activities are affected [2]. The dyes used nowadays are generally cancer-causing and have negative environmental consequences [3]. These dyes are made up of an aromatic chemical and a metal, and their photosynthetic activities are harmful. The majority of mutagenic activities are linked to colors (dye) used in the textile industry [4]. For the treatment of textile wastewater, both physical and chemical methods are used. These procedures include oxidation, membrane technology, flocculation, coagulation, and adsorption, all of which are expensive and may result in secondary contamination as a result of excessive chemical usage. Other less expensive procedures for decolorization include ozonation, electrochemical destruction, and photo catalysis [5]. The most practical way to remove dye is biological therapy, which uses a large number of microorganisms in the declaration and mineralization of a variety of colors. It is quite inexpensive, and the end result of biological treatment is non-toxic. However, because of the limited biodegradability of dyes, there is less flexibility in design and operation [6]. As a result, adsorbents such as activated carbon are employed for dye, although they are not commonly used because of their expensive cost. Peat plum kernels, wood coal resin, and chitosan fiber are some of the adsorbents employed in different industrial pollutants. Few of these adsorbents are widely accessible and inexpensive, but they cannot completely remove dyes such as activated carbon; therefore, it is necessary to develop low-cost adsorbents that may be utilized in place of activated carbon [7,8,9].

To conclude, a molecularly imprinted polymer (MIP) is the best choice for removing dyes from a variety of sources because of its specificity and selectivity, as well as its low cost and simplicity of preparation. Making a molecularly imprinted polymer (MIP) involves combining a template molecule with a functional monomer in the presence of an initiator and a cross linker to generate a polymer that is extremely specific and selective to the template molecules. After washing, cavities comparable in shape and size to the template molecules are generated from the polymerization of monomers and the cavities left in the polymer matrix. Because of their great selectivity even in complicated samples, MIPs used in dyes are utilized as a sorbent for solid phase extraction [10].

The most commercial application of an MIP is in the sample preparation for environmental, food analysis and environmental analysis. Clenbuterol solid-phase extraction (SPE) material is currently available from a Swedish manufacturer [11]. MIPs are popular recognition elements in sensors, and many transducers are employed in conjunction with it [12]. The quartz crystal microbalance, an acoustic transducer sensor, has gained a lot of popularity due to its inexpensive cost and ease of use [13]. The most often utilized universal functional monomer for the preparation of an MIP is methacrylic acid (MAA), and its binding capacity is determined by the bond sites and second pore size of polymeric substances [14]. Ethylene glycol dimethacrylate (EGDMA) is widely utilized as a cross linker, and the cross linker influences the hardness, strength, and selectivity of an MIP. The type and quantity of cross linkers have significant impacts on the polymerization process. If the amount is modest, an unstable polymer will be developed, while a larger amount will lower the number of recognition sites [15]. Acetonitrile, chloroform, dichloroethane, and methanol are among the most often-used solvents for MIP synthesis [16]. The imprinting efficiency, structural adsorption, and interaction between the functional monomer and template will all be affected by the porogen solvent. The use of a less polar porogen solvent promotes the formation of functional-monomer–template complexes, whereas using a more polar porogen solvent disrupts complex interactions [17]. While azo and analogue compounds are applied for the synthesis of an MIP, azobisisobutyronitrile (AIBN) is the best initiator since its decomposition temperature ranges from 50 to 70 °C [18].

There are various methods for MIP preparation, such as suspension polymerization, precipitation, etc., but the most commonly used method for MIP synthesis is bulk polymerization, in which the template is printed in the polymer matrix and the template monomer must be completely removed after polymerization. To convert an MIP to a tiny powder, mechanical breakup and crushing using a mortar and pestle are required [19]. The purpose of this study was to develop an extremely selective MIP adsorbent for acid black-234 (AB-234) dye, as well as to explore the selectivity, rebinding, and use of MIPs in various effluents. The MIP and NIP were produced using bulk polymerization for the rapid determination of AB-234 dye in water samples, but the AB-234 dye showed more selectively toward the MIP than NIP due to the recognition property of the MIP network. The adsorptive mechanism of AB-234 removal by MIP is summarized in Figure 1.

## 2. Results and Discussion

### 2.1. Choice of the Reagents

For the synthesis of a high-affinity molecularly imprinted polymer (MIP) the following conditions are necessary:

Polar porogen solvent and template, high nominal level of cross linker for stable polymer, and one or more functional monomers [20].

### 2.2. Characterization

#### 2.2.1. Characterization by SEM

Scanning electron microscopy (SEM) may be used to determine the size and shape of the MIP and NIP. The use of SEM to analyze the MIP and NIP particles has been reported in several studies [21,22]. For careful consideration of sample morphology and smoothness, SEM is an essential and valued analytical approach. The morphologies of unwashed and washed MIP and NIP samples are shown in Figure 2. The non-covalent precipitation polymerization process of the MIP was responsible for permeability smoothness, with an extremely small, uniform, spherical, and equal size and shape, while no smoothness was observed for the NIP, as shown in the figures. Because the binding kinetic was exposed to the surface, the consistent size of the MIP revealed that the sample enabled the removal of an efficient template. The unwashed sample dye bonded to the polymer without a clear and uniform size and behaved like crystal reagents.

#### 2.2.2. Characterization by FTIR 

FTIR study was carried out within range of 4000–500 cm^−1^ and describes the surface groups of a polymer (MIP) as shown in Figure 3. The starting materials of the MIP and NIP, such as the functional monomer, cross linker, initiator, etc., were the same. Therefore, the overall data of both graphs has an approximate similarity. The peaks at 3400 cm^−1^ and 2900 cm^−1^ were caused by the presence of OH and CH, respectively, whereas stretching at 1720 cm^−1^ and 1200 cm^−1^ was caused by the presence of C=O and C-O, respectively [23,24]. Additionally, the peaks stretching at ~1400 cm^−1^, ~1300 cm^−1^, ~1200 cm^−1^ are caused by the presence of -CH_2_, -CH_3_, and C-O, respectively. The MAA and EGDMA C=C double bond stretching 1637 cm^−1^ peak was absent in the MIP and NIP, indicating that polymerization was successfully carried out.

#### 2.2.3. Brunauer–Emmett–Teller Analysis

The porosity and specific area of a chosen AB-234 molecularly imprinted polymer was determined via BET analysis. The specific surface area of the MIP was 232.321 m^2^/g, with a pore volume of 0.056 cc/g and a pore radius of 12.245 (Å), while the NIP has specific surface area, volume and pore radius 32.034 m^2^/g, 0.0074 cc/g, and 3.441 (Å), respectively as given in Table 1. For the MIP, a greater surface area suggests that unique cavities were generated for AB-234 identification. The MIP has an enhanced adsorption capability due to its larger surface area [25]. The volume ratio of the open pore to total volume is referred to as a particle’s porosity [26]. The template in polymerization has the greatest impact on the surface area and porosity of the MIP. The wider pore of the MIP indicates that the structure of the MIP is not more compact compared to the NIP. The typical pore diameter of the MIP is between 2 and 5 nm, indicating that the polymer is mesoporous [27].

### 2.3. Effect of Adsorbent Mass and pH

The effect of the adsorbent on the adsorption of acid black-234 dye was investigated, and the polymer dosage changed from 2 to 12 mg. The adsorption rapidly increased from 2 to 6 mg at beginning for both the MIP and NIP, but no influence on dye adsorption was observed after 8 mg. Therefore, maximum adsorption (94%) occurred at 8 mg. The pH of the solution was found to range from 2 to 7. It was observed that adsorption was small at a low pH and increased with increasing pH; hence, the maximum adsorption (94%) was recorded at pH 5 as shown in Figure 4. Therefore, this pH was chosen for further study.

### 2.4. Contact Time Effect of AB-234 Dye Adsorption on the MIP and NIP

Kinetics is important to for obtaining information regarding the rate controlling and binding mechanism. Therefore, the contact time was studied as a function of temperature. The effect of contact time (Figure 5) on adsorption was determined at different times (min), while keeping other parameters constant (pH, dose, volume). The adoption was studied at 283K, 298K and 313K, which indicates that the removal of dye is time-dependent for both the MIP and NIP. The results show a gradual increase in the adsorption of dye in the MIP when the contact time increased from 5 to 20 min, followed by a considerable increase in adsorption (94%) at 40 min. After that, no change was observed; therefore, this time was used throughout the study. In general, the dye adsorption process is divided into two phases: a rapid initial sorption phase, followed by a protracted period of relatively slower adsorption [28]. Therefore, initially, uptake of dye was fast, especially during the first 35 min, most likely due to the exposure of most of the binding sites on the MIP.

#### 2.4.1. Pseudo First Order Kinetic Model

A pseudo–first-order kinetic model provides information about the rate of occupied and unoccupied sites, in which different parameters were calculated using the following equation [29]:(1)$$\log\;(q_{e}-q_{t})=\log\;q_{e}-k_{1}\;\frac{t}{2.303}$$

The amounts of AB-234 dye (mg g^−1^) adsorbed at time t (min) and at equilibrium are represented by q_t_ and q_e_, respectively. The pseudo–first-order constant k_1_ (min^−1^) was calculated using a graphing log (q_e_ − q_t_) against “t”. Figure 6a describes a pseudo-first order kinetic model, with various parameters listed in Table 2. The qe (cal) and qe (exp) do not match; therefore, the values show that the adsorption of acid black-234 dye in the MIP does not follow first-order kinetics because the regression coefficient R^2^ = 0.7037 is far from unity.

#### 2.4.2. Pseudo–Second-Order Kinetic Model

Pseudo–second-order kinetic model was used to further analyze the kinetic data. The given equation shows the linear version of pseudo–second-order kinetic model [30].
(2)$$\frac{t}{\mathrm{qt}}=\frac{1}{k_{2}{\;q}_{e}{}^{2\;}}+\frac{t}{q_{e}}$$

The pseudo–second-order constant k_2_ (mg g^−1^ min^−1^) is computed from the plot of t/qt vs. “t” in the above equation. The pseudo–second-order model was used to determine the adsorption of AB-234 dye in the MIP. The R^2^ value for second-order kinetics is larger than the value of pseudo–first-order kinetics, and qe (cal) and qe (exp) have a similar relationship. As a result, we may infer that the pseudo–second-order model’s adsorption data have the best fit. These kinetic data reveal that AB-234 dye adsorption is affected by both the adsorbent and the adsorbate because the regression coefficient R^2^ value 0.9667 is close to unity. The various parameters obtained from this plot are shown in Table 2 and Figure 6b, illustrate that the acid black-234 dye follows pseudo–second-order kinetics.

#### 2.4.3. Intraparticle Model

Weber and Morris claim that, instead of the contact period “t”, the sorption capacity varies according to t^1/2^ Equation (4) contains the linear expression [31].
(3)$$\mathrm{qt}=k_{\mathrm{id}}\;t^{1/2}+\;C$$
where C is intercept and Kid is the rate constant. Multiple stages are involved in the adsorption of dye (AB-234) from an aqueous fluid onto the polymer surface. This process includes the molecular diffusion of sorbate molecules from the bulk phase to the adsorbent outer surface, also known as film or external diffusion. Internal diffusion occurs in the second stage, in which sorbate molecules travel from the MIP surface to the interior locations. The adsorption of sorbate molecules from interior locations to inner pores is the third phase [32].

Intra particle diffusion plot as shown in Figure 6c; however, it fails to pass from its origin due to a difference in the rate of mass transfer between the beginning and final temperatures. Furthermore, such a large divergence from the origin indicates that pore diffusion is not the primary rate control step [33]. When the value of “C” is compared to the rate constant, it is clear that intraparticle diffusion is not just a rate-limiting process, while AB-234 dye adsorption on the MIP is a complicated process governed by surface sorption and intraparticle diffusion.

### 2.5. Binding Isotherm Models of Acid Black-234 Dye Adsorption

The adsorption isotherm provides information about the mechanism of the adsorption process’. The isotherms are used to classify adsorption systems because they demonstrate the adsorption process. The monolayer formation is generally defined by the Langmuir isotherm, and the non-covalent behavior of the MIP is characterized by the Freundlich model [34]. As a result, the Langmuir and Freundlich isotherms were used to assess the adsorption data.

#### 2.5.1. Langmuir Model

The Langmuir model describes a monolayer as an adsorbate surface that is uniform and homogeneous. Adsorption happens at a particular homogenous spot inside the body of the adsorbent. Equation (4) describes the Langmuir model [35]:(4)$$\frac{C_{e}}{Q_{e}}=\frac{1}{K_{L}Q_{m}}+\frac{C_{e}}{Q_{m}}$$
where C_e_ (mg-L^−1^) represents the dye’s liquid-phase equilibrium concentration; Q_m_ (mg-g^−1^) represents the adsorbent’s maximum adsorption capacity; K_L_ (L-mg^−1^) represents the amount of dye adsorbed, the energy or net enthalpy of adsorption; and Q_e_ (mg-g^−1^) represents the quantity of dye adsorbed. C_e_/Q_e_ and C_e_ must have a linear connection with a slope of 1/Q_m_ and an intercept of 1/(Q_m_ KL). Table 3 shows the KL, Q_m_, and R^2^ values derived from the curve (Figure 7a). The MIP has a maximal adsorption capacity (Qmax) 100 mg g^−1^ at 313K. The Langmuir model provides the R^2^ value that best fits the experimental result as given in Table 3.

#### 2.5.2. Freundlich Model

The adsorption characteristics of multilayer and heterogeneous surfaces with uneven adsorption sites and unusually accessible adsorption energy were determined using the Freundlich isotherm. The Freundlich isotherm model equation is shown below [36].
(5)$$\ln\;q_{e}=\ln\;\mathrm{Kf}+\frac{1}{n}\;\ln C_{e}$$
where C_e_ (mg-L^−1^) is the liquid phase concentration at equilibrium, qe mg-g^−1^ is the dye adsorption quantity, Kf (mg-g^−1^) is a relative indication of adsorption capacity, and 1/n is the surface heterogeneity factor indicating adsorption type. For favorable adsorption, the 1/n ratio should be less than 1, whereas for unfavorable adsorption, the value should be larger than 1, indicating poor bond adsorption [37]. Figure 7b depicts the Freundlich model, while Table 3 lists the various parameters.

### 2.6. Thermodynamic Study

Thermodynamic experiments were conducted to examine the dye (AB-234) adsorption process on the MIP. In this study, a significant judgment must be made on whether the mechanism is spontaneous or not. The following equation was used to determine many thermodynamic parameters, including standard free energy (G°), enthalpy (H°), and entropy (S°) [38].
(6)$$\log\mathrm{Kc}=\frac{\Delta S{}^{\circ}}{2.303R}-\frac{\Delta H{}^{\circ}}{2.303\mathrm{RT}}$$
(7)ΔG°=ΔH°−TΔS°
where T is the specific heat, R is the universal gas constant (8.314 J-mol^−1^K^−1^), and K_c_ (Lg^−1^) is the thermodynamic equilibrium constant defined by q_e_/C_e_. The intercept and slope of a plot log K_c_ vs. 1/T were used to calculate the values of ΔS° and ΔH° (Figure 8). Table 4 shows the different thermodynamic parameters that were examined at various temperatures. The value of ΔG° was found to be negative at all temperatures, indicating that dye (AB-234) adsorption in the MIP was spontaneous [39]. The value of ΔG° decreased as the temperature increased, indicating that a higher temperature enhances dye (AB-234) adsorption in the MIP. The positive sign of ΔH° indicates that this adsorption is endothermic, because with the increasing temperature, the rate of adsorbate diffusion on the adsorbent also increased. Additionally, a positive ΔS° value shows that disorder increased during adsorption.

### 2.7. Selectivity Study

A competitive adsorption study was carried out in the presence of dyes, such as AB1, BB3, safranin, and acid yellow 76, that are both comparable and dissimilar in structure to the template (AB-234) to confirm the selective cavities in a polymer. The MIP should be more selective for the specific template (AB-234) than the NIP. At optimum conditions, 100 mg/L of AB2-34 dye and the interfering dyes were added to each Erlenmeyer flask, and the mixture was agitated in a thermostat shaker at 130 rpm. At a certain time, samples were withdrawn, the absorbance was measured using a UV–vis spectrophotometer, and the quantity of dye adsorbed on the polymer was estimated by subtracting the final dye concentration from the starting dye concentration in the mixture. MIP has a 94% selectivity for AB-234, compared to other dyes with adsorption rates ranging from 5% to 23%. Therefore, the selectivity results confirm that MIP cavities are solely selective for the template (AB-234) molecule. The results are graphically shown in Figure 9.

### 2.8. Distribution Ratio and Imprinting Factor

The contact and strength of the template molecule (AB-234) with the polymer (MIP/NIP) is described by the imprinting factor. It appears that the MIP and NIP have recognition properties for a certain analyte. Equation (9) was used to derive the imprinting factor (IF) for a molecularly imprinted polymer:(8)$$\mathrm{IF}\;\propto =\frac{Q_{\mathrm{MIP}}}{Q_{\mathrm{NIP}}}$$
where Q_MIP_ denotes MIP adsorption capacity for the dye (AB-234), and Q_NIP_ denotes NIP adsorption capacity for AB-234. The following equation was used to calculate the distribution ratio.
(9)$$K_{d}=\frac{\left(C_{i}-C_{f}\right)}{C_{f}\;m}V$$
where Kd (L/g) denotes the distribution coefficient, C_i_ the initial dye concentration, C_f_ the final dye concentration, V the volume used, and “m” the polymer mass (MIP/NIP) [40]. The obtained results have been summarized in Table 5.

### 2.9. Application of MIP to Adsorb Dye (AB-234)

The polymer was applied as a solid-phase adsorbent material in the preconcentration of AB-234 from river water and effluent samples in order to test the efficiency of the MIP produced in real environmental samples. In a total volume of 10 mL, aliquots of these three samples were spiked with known volumes of AB-234 dye standard solution at concentrations of 50, 75, and 100 mg/L, with agitation for 40 min. Given the MIP’s highest retention capability of 94.67% (±0.1) while the recoveries were between 84 and 94%, as shown in Table 6. Taking into account the complexity of the examined samples, these results reveal that the imprinted material shows an outstanding sorption capability and ability to give unique recognition of the analyte.

### 2.10. Reuse of MIP for AB-234 Dye

The continuous use of the MIP was determined in five cycles of adsorption–desorption under optimal conditions. Table 7 shows that, between the first and fifth cycles, roughly 14% of the AB-234 dye rebinding was lost. According to the results, the MIP could be reused at least five times without significantly lowering its adsorption capability.

A comparison present adsorbent capacity with those cited in literature in given in Table 8.

## 3. Materials and Methods

### 3.1. Materials

All solvents and dyes were analytical-grade and were supplied by Sigma-Aldrich (Taufkirchen, Germany). The acid black-234 dye (AB-234) as a template, methacrylic acid (MAA) as a functional monomer (Dae-Jung, Korea), azobisisobutyronitrile as a reaction initiator, methanol as a porogenic solvent (Chengdu, China), acetone and methanol mixing solution as a washing solvent (Merck, Darmstadt, Germany), and ethylene glycol dimethacrylate (EGDMA) as a cross linker (J.T. Baker, New York, NY, USA). The MIP’s selectivity for AB-234 was tested using AB1, BB3, safranin, and acid yellow-76. The water was deionized using a Milli-Q system (Chennai, India). The chemical structures of the dyes are shown in Figure 10.

### 3.2. Characterization of the MIP and NIP

The structure of polymer was studied using FTIR in the range of 4000–500 cm^−1^; Vertex 70 (Shimadzu, Kyoto, Japan) and SEM (JSM-IT500) were used to assess the size and shape of the MIP and NIP; and a UV–Vis 1800 spectrophotometer was used to quantify absorbance using quartz cuvettes (Shimadzu, Japan). Brunauer–Emmett–Teller theory was used for the surface area calculation from the adsorption of nitrogen (8 mg MIP and NIP, N2, 70 °C, 5 h) (ASAP 2010).

### 3.3. Preparation of the MIP and NIP

#### Preparation of the MIP for AB-234 Dye

For the synthesis of the MIP, 0.215 g of acid black-234 dye was dissolved in 10 mL of methanol and stirred for 10 min. Then, 150 mmol of MAA was added and left to rest for two hours, followed by the addition of 225 mmol of EGDMA, and then left for 15 min. Then, 2 mg of initiator (ABIN) was added and maintained in a water bath for 24 h at 60 °C. The NIP was synthesized using same procedure but without dye. The sample was removed from the flask after 24 h of heating, and the polymer was filtered. After filtering, the polymer was washed five to six times in a soxlet system with a methanol/acetone solution (8:2, *v*/*v*) to completely remove the template. Finally, a pure polymer was produced, which was then dried at room temperature. The synthesis protocols of the MIP and NIP are shown in Table 9.

### 3.4. Binding Adsorption Analysis

MIP binding was studied using 20 mL vials containing 8 mg of an MIP and 10 mL of 100 mg/L dye by adjusting parameters, such as mass, concentration, pH, and time. After 40 min on a centrifuge at 15,000 rpm, the supernatant was filtered through a 0.45 µm membrane before UV–Vis spectrophotometric measurement. The following equation was used to calculate the binding adsorption capacity:(10)$$Q=\frac{\left(C_{0}-C_{e}\right)}{V}m$$

The initial dye concentration is Co (mgL^−1^), the equilibrium dye concentration is Ce (mgL^−1^), the experimental adsorption quantity is Q (mg-g^−1^), the volume of solution is V, and the mass of the MIP is m (g).

### 3.5. Selectivity Study

A competitive adsorption study was carried out in the presence of molecules that are similar and different in structure to AB-234 dye in order to assess the creation of selective cavities in polymer. Different dyes were used in this case, including acid black-1 (AB-1), safranin, acid yellow-76 (AY-76), and BB-3. For each compound’s selectivity recognition assays, 8 mg of the MIP was dissolved in a 10 mL solution (pH 5), containing 100 mg/L of each dye and equilibrated for 40 min. After stirring the concentration of these dyes in the supernatant was determined using a UV–Vis spectrophotometer.

## 4. Conclusions

The current study looks at how to employ a synthetic polymer (MIP) to remove a specific analyte (AB-234) from various water samples under optimal conditions. Compared to other dyes—acid black-1 (AB1), acid yellow-76 (AY-76), safranin, and BB-3, which had 23%, 18%, 9.3%, and 7% selectivity, respectively, toward a specific analyte— AB-234 had about 94% selectivity toward a specific analyte. The adsorption of AB-234 on NIP was only 31%, which indicates that the MIP is more specific for AB2-34 dye due to the formation of complementary cavities. The MIP followed second-order kinetics and the Langmuir model, according to kinetic and isotherm analyses. Therefore, a high adsorption (94%) was found on 313 K. The negative value of ΔG° shows the process to be spontaneous, while the ΔH° and ΔS° demonstrates the endothermic and feasible nature of the process. The MIP can be easily and repeatedly be recovered from the solution using the centrifugation process, without the immense loss of its selectivity. Therefore, the MIP synthesized here could be presented as a possible material for the separation of AB-234 dye from wastewater with immense selectivity and great recovery.

## Figures and Tables

**Figure 1 molecules-28-01555-f001:**
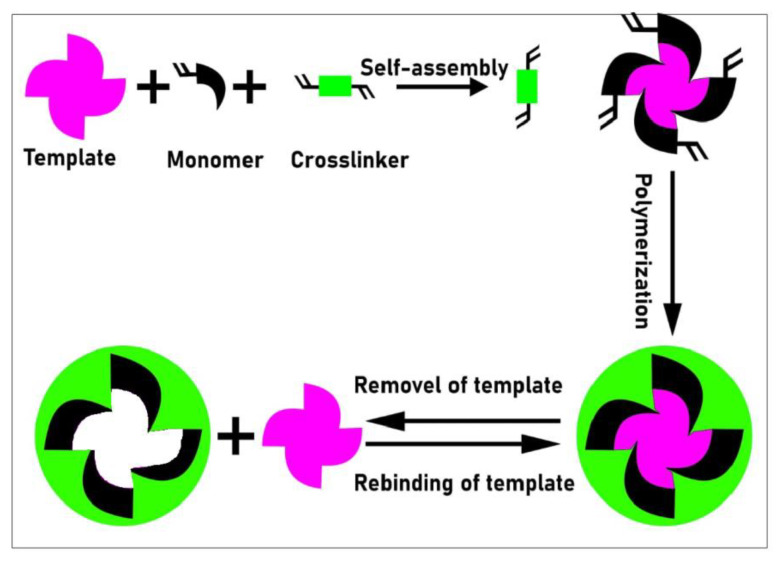
Adsorptive mechanism scheme of AB-234 dye on an MIP.

**Figure 2 molecules-28-01555-f002:**
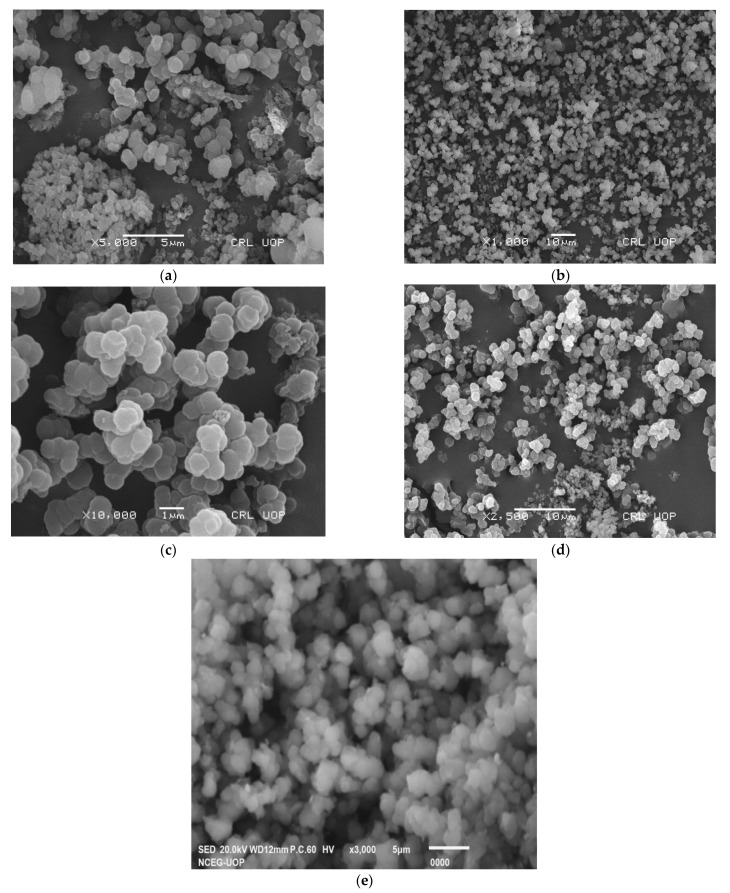
SEM micrographs after adsorption (**a**,**b**) and washing (**c**,**d**) of an MIP and (**e**) NIP.

**Figure 3 molecules-28-01555-f003:**
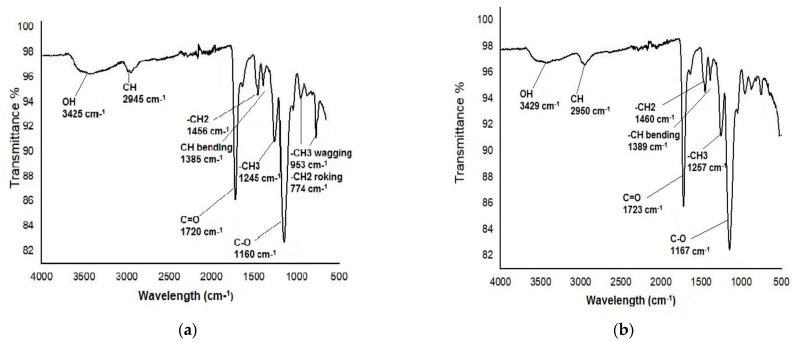
FTIR spectra of the MIP (**a**) and NIP (**b**).

**Figure 4 molecules-28-01555-f004:**
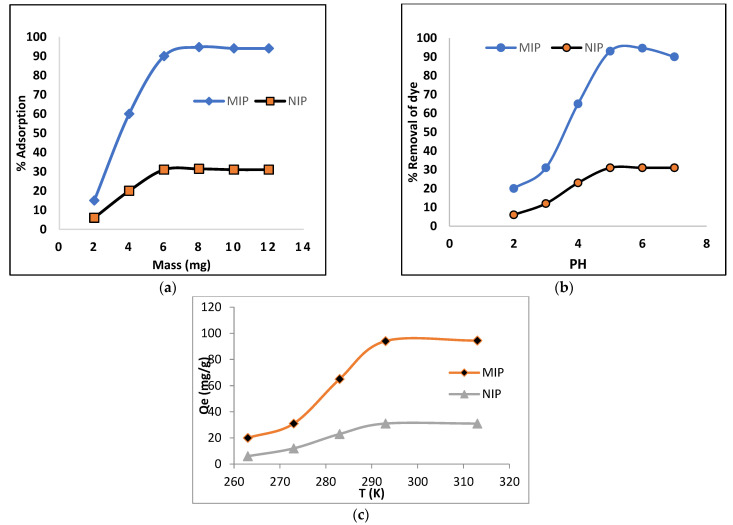
Effect of mass (**a**), pH (**b**) and temperature on adsorption of AB-234 dye on the MIP and NIP (**c**).

**Figure 5 molecules-28-01555-f005:**
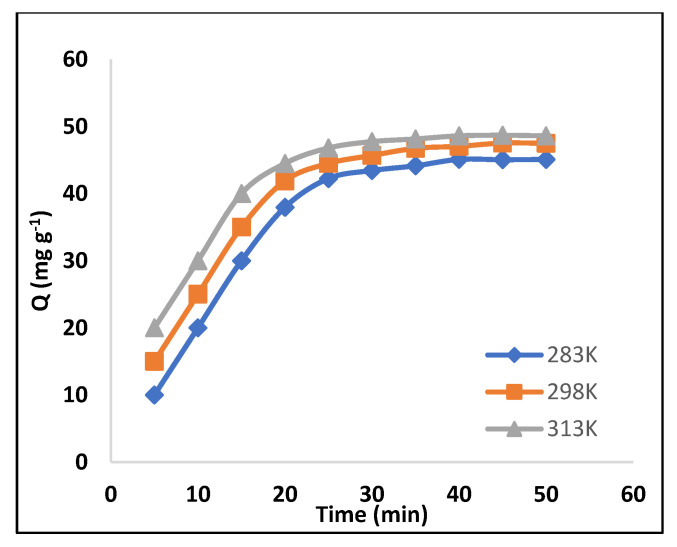
Adsorption kinetics of AB-234 dye in the MIP, polymer mass 8 mg, V = 10 mL, pH = 5, AB-234 = 100 mg/g.

**Figure 6 molecules-28-01555-f006:**
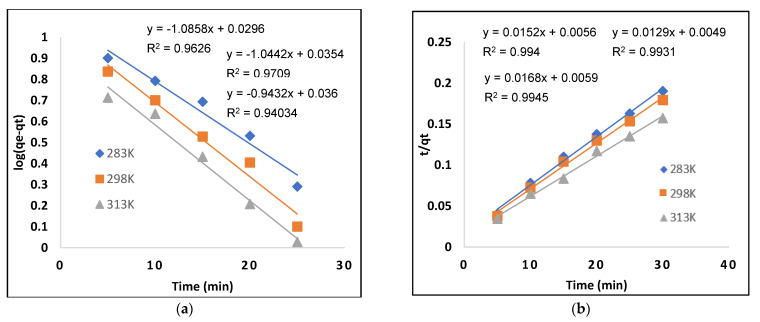
Kinetic model for AB-234 dye adsorption: (**a**) pseudo–first- and (**b**) pseudo–second-order kinetics (**c**) intraparticle models.

**Figure 7 molecules-28-01555-f007:**
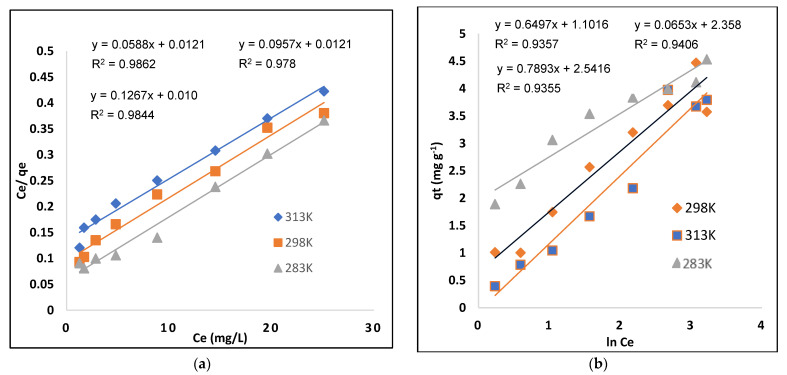
Isotherm models: (**a**) Langmuir and (**b**) Freundlich.

**Figure 8 molecules-28-01555-f008:**
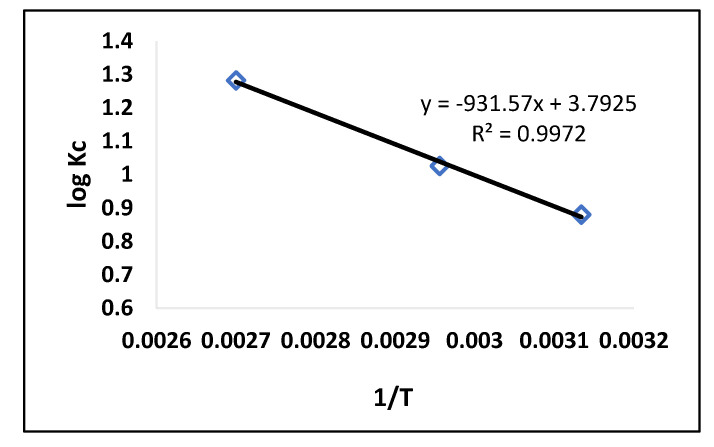
Van’t Hoff graph (MIP).

**Figure 9 molecules-28-01555-f009:**
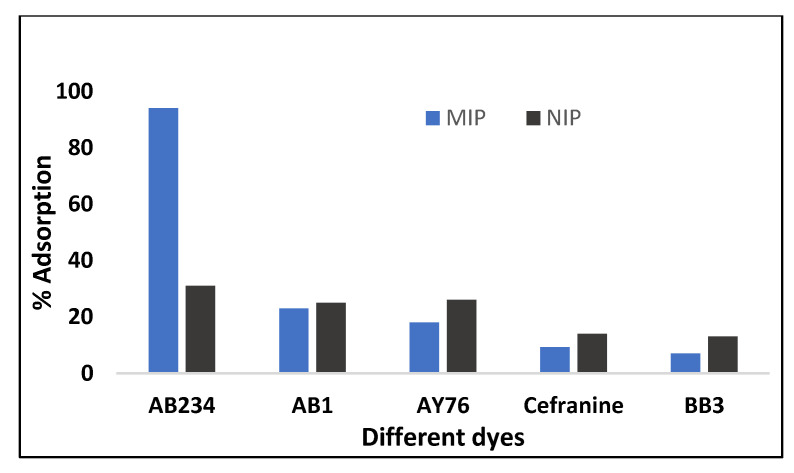
Selectivity of MIP towards different similar structures dyes.

**Figure 10 molecules-28-01555-f010:**
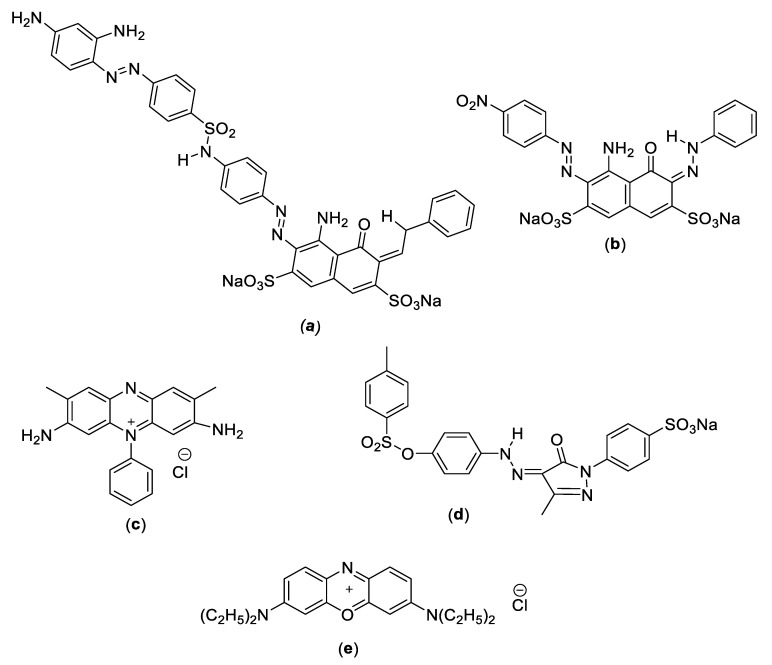
Structure of the different dyes (**a**) acid black-234 (AB234) (**b**) acid black-1 (AB1) (**c**) safranin (**d**) acid yellow-76 (AY76) (**e**) basic blue-3 (BB3).

**Table 1 molecules-28-01555-t001:** BET analysis of MIP.

Polymer	Specific Surface Area (m^2^/g)	Pore Volume (cc/g)	Pore Radius (Å)
MIP	232.321	0.056	12.245
NIP	32.034	0.0074	3.441

**Table 2 molecules-28-01555-t002:** Different kinetic models’ parameters for the adsorption of AB-234 dye on MIP.

Parameters	283 K	c	313 K
Pseudo first-order			
K1	0.0643	0.0729	0.0841
Qe (calculated)	9.342	11.543	14.768
Qe (experimental)	41.654	42.545	42.653
R^2^	0.9626	0.9709	0.9799
Pseudo second-order			
K2	0.0069	0.0127	0.0141
Qe (calculated)	43.785	44.654	44.876
Qe (experimental)	41.654	42.545	42.653
R^2^	0.9945	0.994	0.9783
Intraparticle diffusion			
K_id_ (mg/g min^−1/2^)	3.04	2.32	1.96
C	28.87	34.38	37.083
R^2^	0.9764	0.9508	0.9683

**Table 3 molecules-28-01555-t003:** Parameters of Langmuir and Freundlich models for the adsorption of AB-234 on the MIP.

Parameters	283 K	298 K	313 K
Langmuir model			
Qm (mg-g^−1^)	82.23	83.04	100
KL	0.207	0.125	0.789
R^2^	0.9862	0.978	0.9844
Freundlich model			
Kf (mg-g^−1^) (L mg-g^−1^)	3.008	10.543	12.68
1/n	0.6497	0.653	0.7893
R^2^	0.9358	0.9406	0.9355

**Table 4 molecules-28-01555-t004:** Different thermodynamic parameters for AB-234 dye.

Temperature (K)	ΔG° KJmol^−1^	ΔH° KJmol^−1^	ΔS° KJmol^−1^
288	−1439	7.74	31.52
298	−1915		
308	−2516		

**Table 5 molecules-28-01555-t005:** Adsorption parameters for dyes (MIP/NIP).

Dyes	% Removal		Adsorption Capacity Q mg/g		Distribution Coefficient Kd (L/g)		Imprinting Factor $\mathrm{IF}\;\propto =\frac{Q_{MIP}}{Q_{NIP}}$	Selectivity $S=\frac{I^{AB234}}{I^{\;interfering}}$
	MIP	NIP	MIP	NIP	MIP	NIP		
AB-234	94	31	41.1	8.0	0.53	0.06	5.13	-
AB-1	23	25	5.21	6.31	0.42	0.27	0.82	6.25
SEFRANIN	18	26	4.32	6.61	0.02	0.01	0.65	7.89
AY-76	9.2	14	2.35	4.30	0.21	0.15	0.54	9.50
BB-3	7	13	1.38	3.15	0.17	0.11	0.43	11.9

**Table 6 molecules-28-01555-t006:** Recovery test of the MIP using different samples in a province of Pakistan (KPK).

Samples	Added (mg/L)	Found in MIP (mg/L)	Recovery (%)	RSD (%)
Textile industry effluent	50	46.51 ± 0.04	93.02	0.33
	75	71.02 ± 0.07	94.60	0.34
	100	94.8 ± 0.1	94.8	0.39
River 1	50	45.43 ± 0.03	90.86	0.30
	75	71.05 ± 0.1	94.66	0.54
	100	86.62 ± 0.09	86.62	0.77
River 2	50	43.62 ± 0.02	87.24	0.04
	75	66.54 ± 0.1	88.01	0.49
	100	84.89 ± 0.04	84.89	0.61

**Table 7 molecules-28-01555-t007:** Repeated use of MIP on adsorption capacity.

**Times**	1	2	3	4	5
**Q (mg g^−1^)**	94	92	89	84	80

**Table 8 molecules-28-01555-t008:** Adsorptions capacity of acid black-234 dye on different adsorbents.

Adsorbents	Adsorption Capacity (mg-g^−1^)	Ref
Polyaniline/chitosan (PAn/Ch)	74.1	[32]
Starch (PPy/St)	62.5	[41]
Polyanaline/sugarcane begasse (PAn/SB)	52.6	[41]
MIP	100	Current study

**Table 9 molecules-28-01555-t009:** Polymerization mixture of MIP synthesis.

Reagents	Chemicals	MIP and NIP Composition (Mass and Volume)	
Template	AB-234	MIP	NIP
Monomer	MAA	0.215 g	-
Cross Linker	EGDMA	150 mmol	150 mmol
Solvent	Methanol and Acetone	225 mmol	225 mmol
Initiator	AIBN	10 mL	10 mL
Template	AB-234	2.00 g	2.00 g

## Data Availability

The raw/processed data required to reproduce these findings cannot be shared at this time due to technical or time limitations.

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
