# Peer review of "Synthesis and Characterization of MIPs for Selective Removal of Textile Dye Acid Black-234 from Wastewater Sample"

_molecules, 2023, doi:10.3390/molecules28041555_

Round 1
Reviewer 1 Report
1. Include the results of adsorption mechanisms in conclusion.
2. The resolution of figures 1 and 3 is low, please provide high-resolution pictures.
3. In order to show the mechanism more clearly, it is better to provide an adsorptive mechanism scheme.
4. Authors need to carefully proofread the grammar for the final version.
5. “The peaks round about 3400 184 cm-1 and 2900 cm-1 were due to the presence of OH and CH, respectively.” This should be updated refs, such as Inorganics, 10(2022) 202 and Micropor. Mesopor. Mat, 341(2022) 112098. “Few of these adsorbents are widely accessible and inexpensive, but they cannot completely remove dyes like activated carbon, thus it is necessary to develop low-cost adsorbents that may be utilized in place of activated carbon” some updated refs could be cited, such as Mater. Today. Commum., 2022, 31,103514 and Chem. Eng. J., 2022, 433, 133857 and Coord. Chem. Rev., 445(2021) 214074
6. I want to check how about the recycle effect for this material?
7. Please provide the SEM after adsorption test.
8. I suggest the authors have to compare similar work on the adsorbing Dye Acid Black-234.
Author Response
Reviewer 1
. Include the results of adsorption mechanisms in conclusion.
Ans: Adsorption mechanism has been included in conclusion.
- The resolution of figures 1 and 3 is low, please provide high-resolution pictures.
Ans: The resolution of mentioned figures has been increased.
- In order to show the mechanism more clearly, it is better to provide an adsorptive mechanism scheme.
Ans: An adsorptive mechanism scheme has been provided on as figure 2.
- Authors need to carefully proofread the grammar for the final version.
Ans: Proofreading has been done.
- “The peaks round about 3400 184 cm-1 and 2900 cm-1 were due to the presence of OH and CH, respectively.” This should be updated refs, such as Inorganics, 10(2022) 202 and Micropor. Mesopor. Mat, 341(2022) 112098. “Few of these adsorbents are widely accessible and inexpensive, but they cannot completely remove dyes like activated carbon, thus it is necessary to develop low-cost adsorbents that may be utilized in place of activated carbon” some updated refs could be cited, such as Mater. Today. Commum., 2022, 31,103514 and Chem. Eng. J., 2022, 433, 133857 and Coord. Chem. Rev., 445(2021) 214074.
Ans: The references as suggested by reviewer has been inserted in the text as reference 9 and 24.
- I want to check how about the recycle effect for this material?
Ans : The recycle effect is already mentioned as section 3.10 and the result are tabulated as table 8.
- Please provide the SEM after adsorption test.
Ans: SEM images after adsorption has been mentioned in figure 3
- I suggest the authors have to compare similar work on the adsorbing Dye Acid Black-234.
Ans: A comparison table has been added in text as table 9
Reviewer 2 Report
1. In the abstract, the “MIP” term must be defined
2. Table 1. “Polymerization mixture of MIP synthesis” must be redesign to concord with provided data
1. The subchapter “3.1 Choice of the reagents” must be reformulated for a better understanding
2. In chapter 3.2.1. “Characterization by SEM” the authors state that “SEM analyse MIP and NIP particles has been reported in several research studies” but reported a single reference. Please clarify
4. For a better comparison, the SEM images form figure 2 must be at the same magnification
5. In figure 2. which images are MIP and which are NIP? Please specify.
6. The FT-IR of MIP and NIP after dye absorbance and before washing was performed? Are any differences in respect to raw polymers?
7. In figure 6.b and 6.c the temperature legend for all value must be presented and be the same
8. For the figure 6 and figure 7 the shape, colour and temperature legends should be the same for a better understanding and consistency
9. In all the paper, the subscript and superscript of indexes should be respected according to used formula
Author Response
Reviewer 2
In the abstract, the “MIP” term must be defined.
Ans: MIP term has been defined in abstract part of the manuscript.
- Table 1. “Polymerization mixture of MIP synthesis” must be redesign to concord with provided data.
Ans; Table 1 has been redesigned.
- The subchapter “3.1 Choice of the reagents” must be reformulated for a better understanding.
Ans: Choice of the reagents has been reformulated in section 3.1.
- In chapter 3.2.1. “Characterization by SEM” the authors state that “SEM analyse MIP and NIP particles has been reported in several research studies” but reported a single reference. Please clarify.
Ans: Another reference has been added in this section,(ref 22)
- For a better comparison, the SEM images form figure 2 must be at the same magnification
Ans: We should have performed SEM analysis at the same magnification but unfortunately due to some fault in SEM equipment we are not able to take the images again.
- In figure 2. which images are MIP and which are NIP? Please specify.
Ans: NIP image has been included in the manuscript as Fig 3 (e)
- The FT-IR of MIP and NIP after dye absorbance and before washing was performed? Are any differences in respect to raw polymers?
Ans: Unfortunately the FTIR spectra of MIP and NIP after dye absorption were not carried out due to limited resources. Worthy reviewer, we here perform these types of analysis on payment as the instrument are not available everywhere. For each analysis we pay and would have to wait for months as well.
- In figure 6.b and 6.c the temperature legend for all value must be presented and be the same.
Ans: The temperature used was the same and it has been presented in respective figures.
- For the figure 6 and figure 7 the shape, colour and temperature legends should be the same for a better understanding and consistency.
Ans: Correction has been made in mentioned figures
- In all the paper, the subscript and superscript of indexes should be respected according to used formula.
Ans; Correction if any was made.
Round 2
Reviewer 1 Report
can be accepted now